# Sustainable Development of the Rural Areas from Romania: Development of a Digital Tool to Generate Adapted Solutions at Local Level

Eduard Alexandru Dumitru [1], Ana Ursu [2], Valentina Constanta Tudor [1,*] and Marius Mihai Micu [1]

[1]   Faculty of Management and Rural Development, University of Agronomic Sciences and Veterinary Medicine, 010961 Bucharest, Romania; dumitru.eduard@iceadr.ro (E.A.D.); micu.marius@managusamv.ro (M.M.M.)
[2]   The Agricultural Economics Office, Research Institute for Agriculture Economy and Rural Development, 010961 Bucharest, Romania; ursu.ana@iceadr.ro
*   Correspondence: tudor.valentina@managusamv.ro

**Abstract:** Rural Romania faces many problems, the main one being depopulation. In general, young adults frequently choose to migrate to urban centers or leave the country entirely to satisfy economic and/or social aspirations. The aim of this study is to identify intractable issues that inhibit rural development and to identify actionable solutions. In this regard, a questionnaire was developed and administered at the local level. The results obtained were analyzed with the SPSS Statistical Program, with the Pearson Chi-square, and Cramer and Pearson coefficients were determined. The answers obtained from applying the questionnaire made it possible to identify the causes that inhibit the sustainable development of rural areas. It can be considered that a solution that could lead to the sustainable development of the rural areas from Romania would be represented by the development of a digital tool that would create a synergy between local authorities, researchers and the private sector. In order to be able to solve these problems, the Government of Romania should provide the necessary funds to local authorities, depending on the needs identified through the digital instrument, acting directly on the aspects that restrain the development of those areas.

**Keywords:** rural development; regional governments; rural area; localities; Romania

## 1. Introduction

The rural area is an area of vital importance to the European Union, being a priority among the funding measures that the EU makes available to the Member States. Due to a significant share of the population living in rural areas, but also the fact that the activity of basic food production takes place in rural areas, this space needs to be continuously adapted to cope with these changes. Additionally, the rural environment represents the identity of each people, being the space where folk traditions and customs are still preserved and transmitted from generation to generation [1–4].

The rural environment of Romania has undergone numerous changes, being, today, extremely important from the perspective of the significant share of people living in these areas. The changes in the rural areas were highlighted with Romania's accession to the structures of the European Union, at which point the rural environment began to benefit from a series of specific programs and support measures. In Romania, especially in regions located in the lowlands, where the main activity is represented by agriculture, non-agricultural activities as rural tourism are difficult to develop in the absence of natural resources that could attract a large number of tourists. Even when these resources are present and have the potential to attract tourists, poor transport infrastructure makes it difficult to access these areas [5,6].

The village is directly related to the practice of agriculture and the presence of crafts, specific to each region. The two National Rural Development Programs, which Romania

carried out in the period 2007–2013 and 2014–2020, brought important changes to the level of rural localities in Romania [7–9].

Regardless of the stage of development of an area, the rural environment contributes to the economic and social development of the area, in the region in which it is located. The countryside is more than a source of labor for agriculture, and a supplier of raw materials, the countryside provides jobs for the upstream and downstream sectors of agriculture. Additionally, in rural areas, basic food products are obtained through the rational use of natural resources and the protection of biodiversity [10–12].

In times of health crisis (such as the COVID-19 pandemic), the rural environment can be a good alternative to large urban agglomerations. Thus, this crisis has led to a migration of people from large urban centers to villages and communes located near the cities [13,14].

The rural area can be considered the keeper of traditions and crafts. The Romanian people are internationally recognized, due to the many crafts that are still preserved in the Romanian village. Additionally, through more environmentally friendly agricultural practices, the people from the villages managed to preserve the specific flora and fauna. Natural landscapes and biodiversity contribute to the development of tourism and agro-tourism in rural areas. Therefore, the rural area is not only agricultural, but also allows the development of non-agricultural businesses, and involves a multiplier effect that can contribute to achieving the well-being of rural communities [15–18].

Regarding the rural area, the concept of sustainable rural development has recently emerged. Initially, this concept of sustainable development was mainly focused on the sustainable development of agricultural activities, later sustainable development was extended to rural communities as a whole, taking into account all aspects of geographical, natural, demographic, economic and social to allow the development of all aspects relating to the rural communities. Sustainability is based on the integration of all factors that can contribute to the long-term development and stability of the rural area [19].

The principles of rural development cannot be applied uniformly throughout a country, but they must be implemented locally, after identifying the strengths and weaknesses of the community. In this sense, local authorities play an important role in the process of sustainable development for the areas they represent [20,21].

The rural regions in Romania have not benefited from uniform development, not all areas in Romania are in the same stage of development, there are more developed areas and areas where important investments are needed to ensure sustainable development. The development stage of the rural environment, from the developing regions of Romania, is not uniform. There are rural areas that have developed more harmoniously, due to the access of the European funds, just as there are development regions where the rural environment is in decline, facing an aging population and a high share of agricultural business, as well as the existence of a small number of young people, who return to the village after completing their studies. These differences encountered at the level of the eight development regions of Romania should be balanced by investments that take into account the real needs of the inhabitants, as well as the potential for the development of that area. The main reasons that determined a low percentage of absorption of funds for rural infrastructure development are the lack of funding, but also the absence of specialized staff with experience in the process of accessing European funds. This is also highlighted by the rate of absorption of European funds, where through the Regional Operational Program, the rate of absorption was 44.29% of the total allocations (2014–2020) of 6.9 billion EUR. Furthermore, the absorption rate of programs with a source of European funding, in most cases, does not exceed 60%, by mid-2021 [22–24].

Additionally, in the case of rural areas that are located near urban centers, there can be a positive influence on the development of these areas, first of all, because residents have access to jobs, and second of all because they can invest in infrastructure, more easily, by expanding the existing infrastructure in urban centers [25].

The rural area is of great importance worldwide, being studied by many authors. The modeling of social and economic systems can be a tool to understand the real issues,

identify patterns and develop scenarios. A large part of the existing models in different countries is based on the idea of the functionality of four major agricultural modes: large farms, businesses with various organizational and legal grounds, but also individual households based on different types of savings [26].

The modeling of rural development areas has its own specificity, as it simultaneously combines not only the economic part but also the social, demographic, environmental and other particularities. "Multifunctionality" conditions the achievement of various functions by rural areas, on the other hand, it conditions the diversified development of rural areas [26].

Research made by various authors suggests that the development of rural areas should not be monofunctional, i.e., focus on a particular industry. Additionally, the structure of the rural economy must allow the introduction of new businesses in these areas. The social sphere can have considerable potential for diversifying new types of economic activities. At the same time, diversification not only stimulates economic development but also ensures the social development of rural areas in accordance with the following socially-oriented criteria: employment rate in rural areas, average monthly salary, the standard of living [27].

Although more than 30 years have passed since the fall of the communist regime, the Romanian rural area is in continuous degradation. With Romania's accession to the European Union in 2007, the funds intended for rural development are not found in the field, in the sense that it was not possible to solve the problems of an economic nature (creating a sufficient number of jobs to meet requirements, the development of municipal and sanitary infrastructures) and neither social (stopping the migration of young people). Normally, local authorities should know best what are the aspects that inhibit the development of the localities they run. In this respect, based on the questionnaire applied to the representatives of the local authorities, the main problems they face were identified. The aim of this study is to identify unresolved issues that inhibit the development of rural areas, by creating a "digital" link between local authorities, researchers and the private sector, thus creating a synergy between the three key factors for the development of Romanian rural areas. These results can be extended to other countries in the world, which face similar problems, to act in regions/localities that need funding according to the identified needs.

## 2. Theoretical Background

A model of sustainable development of rural areas was studied in the article "*Problems and Mechanisms of Sustainable Development of Rural Areas*". This model involves the functioning of social infrastructure, improving living standards, establishing an institutional environment, developing and diversifying agricultural activities by stimulating small businesses, crafts, tourism and large-scale development of agricultural cooperatives [28].

The activity of organic production is considered to have a beneficial contribution in terms of rural development. In his article "*Organic Farming and Rural Development: Some Evidence from Austria*", the Austrian researcher pointed out that organic farming can be a solution for rural development, starting from the fact that such organic products are in high demand among consumers, and they are even willing to pay a higher price to eat cleaner, healthier [29].

The contribution of tourism and agrotourism to rural development was also studied. Thus, the authors in the paper entitled "*The model of rural tourism development in Eastern Croatia based on the example of Austria*" developed a model of rural tourism for Croatia inspired from Eastern models of rural tourism existing in Austria. According to the authors, the development model of rural tourism in Eastern Croatia can help to eliminate the problem of regional tourism disparities and bring many benefits to the members of rural communities in Eastern Croatia [30].

Through the paper "*The Bottom-Up Development Model as a Governance Instrument for the Rural Areas the Cases of Four Local Action Groups (LAGs) in the United Kingdom and in Italy*" the authors comparatively analyzed the evolution of rural development policies and

local action groups in the United Kingdom and Italy. The purpose of the paper was to ensure that the partnership approach has given the rural development actors a governance platform to help increase beneficial interactions and economic activity in each of these LAGs, but it is the bottom-up leadership of key local actors, seizing opportunities provided by the EU funding, which have been the most important factors for the LAG successes [31].

Although the rural regions of Romania face many problems, they are accentuated by events that occur cyclically, as was the case of the economic crisis of 2008. In the paper "*Urban or Rural: Does It Make a Difference for Economic Resilience? A Modeling Study on Economic and Cultural Geography in Romania*" the authors identified an interesting change in the regional economy: some economic activities in large urban areas in Romania have moved to nearby rural areas. The article researched and measured the economic resilience of local communities in Romania [32].

The paper entitled "*Development of qualitative monographic studies of rural areas in Germany and Austria*" conducted a study at the level of villages located in Germany and Austria in order to identify the current stage of development of these regions and to identify commonalities as well as good practices in the sustainable development of rural communities in Germany and Austria. This study is relevant as the two areas: Germany and Austria are similar in terms of landforms and are in a similar stage of development [33].

One of the most difficult problems encountered by the management of social infrastructure development in different countries of economic development is the search for CSR management interactions at the national, regional and local level (municipal, neighborhood). In the paper entitled "*The Model of Integrative Management of Rural Social Infrastructure Development*", the authors had as main subject the creation of an integrated model for managing the social infrastructure, based on analysis of rural social infrastructure management (CSR) processes [34].

## 3. Methodology

The purpose of the questionnaire was to identify the economic and social problems faced by local authorities in rural areas, in order to find a solution to address these shortcomings. The study was carried out at the level of the South-Muntenia development region and the West region, the result obtained being able to be applied at the national level as well. This region was selected for analysis because it has the highest share of agricultural production values in all regions of Romania, and most of the workforce is employed in agriculture. Moreover, the natural or anthropic resources that can attract tourists in these regions are extremely few, so that tourism cannot be an alternative for socio-economic development in this region.

The western region includes four counties (Timis-TM, Arad-AR, Caras-Severin-CS and Hunedoara-HD), totaling 357 communes, while in the South-Muntenia region there are seven counties (Arges-AG, Prahova-PH, Dambovita-DB, Teleorman-TR, Giurgiu-GR, Ialomita-IL and Calarasi-CL) which totals 519 communes (Figure 1). To this end, all communes were identified and the questionnaire was sent to them. The number of respondents was 272, out of a total of 876, so it can be stated that the sample is representative, with an error of ± 5%. For the sample to be representative, the minimum number of respondents should have been 267 localities. The questionnaire, consisting of a number of 12 questions, was sent using the e-mail address of the town halls, and their answers were counted with the Google Forms application. It should be mentioned that the data obtained/centralized complies with the legislation in force regarding personal data.

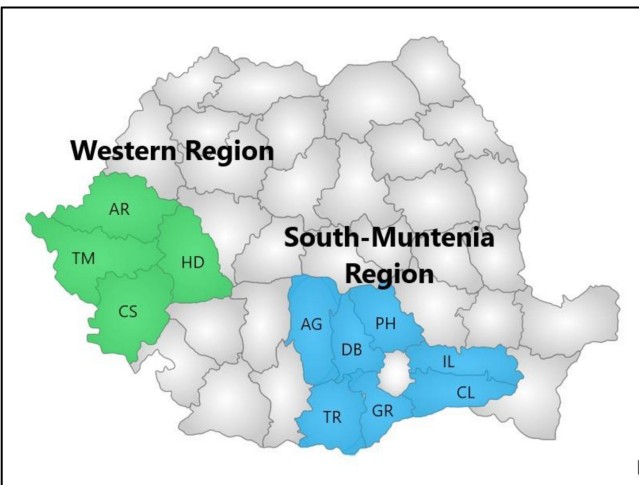

**Figure 1.** Graphic representation of the West and South Muntenia region. Source: in-house development.

**Hypotheses 1 (H1).** *In rural areas, the predominant profile of economic agents is agricultural.*

**Hypotheses 2 (H2).** *Low absorption rate of European funds.*

**Hypotheses 3 (H3).** *Lack of co-financing is the main factor preventing local authorities from accessing European funds.*

**Hypotheses 4 (H4).** *Local authorities do not have qualified staff to identify and access European-funded programs.*

The data collection was carried out between 15 January and 15 February 2021, being applied to local authorities located in the developing regions of West and South Muntenia. The data collection method was performed digitally, the filling out was unassisted. The questions asked were closed and allowed a single answer to be selected. The response rate was 100%. All data were collected with the consent of the participants.

Data processing in terms of descriptive statistics was performed using Pearson Chi-Square, Pearson'R, Cramer V coefficients and the critical value (to highlight the associations between variables), using the SPSS software (SPSS Statistics 20, IBM Software Group, Chicago, IL, USA).

Following the processing of the obtained data, it can be observed that the structure of the respondents according to the number of inhabitants in the locality consists of:

- 42.28% have over 3000 inhabitants,
- 30.51% have between 2001–3000 inhabitants,
- 18.75% have between 1001–2000 inhabitants,
- 8.46% have up to 1000 inhabitants.

Following the processing of the data obtained, it can be seen that the structure of the respondents, depending on the number of villages that make up the analyzed communes, is as follows:

- 40.07% over 3 villages,
- 26.10% over 2 villages,
- 20.96% over 3 villages,
- 12.87% over 1 sat.

As a result of the processing of the obtained data, it can be observed that the structure of the respondents according to the surface of the locality on which is the following:

- 39.71% over 100 km$^2$,
- 34.93% between 50–100 km$^2$,
- 25.37% under 50 km$^2$.

Following the processing of the obtained data, it can be observed that the structure responds according to the dominant profile to the economic agents from the locality presented by the following:

- 83.09% agricultural,
- 16.91% non-agricultural.

## 4. Findings and Discussion

### 4.1. Identifying Social and Economic Problems

Considering the answer given by the representatives of the localities regarding the factor that contributes the most to the current stage of development of the rural community, it can be seen that, for most of the respondents (47.79%) the aging of the population is the main problem. Identified at the level of the community they represent 29.41% of respondents consider that the migration of young people to urban centers is the main problem facing the rural community they represent. In a small percentage, respondents consider that the poor integration of ethnic groups, as well as the existence of a large number of socially assisted people, are among the main problems of the communities they represent (Figure 2).

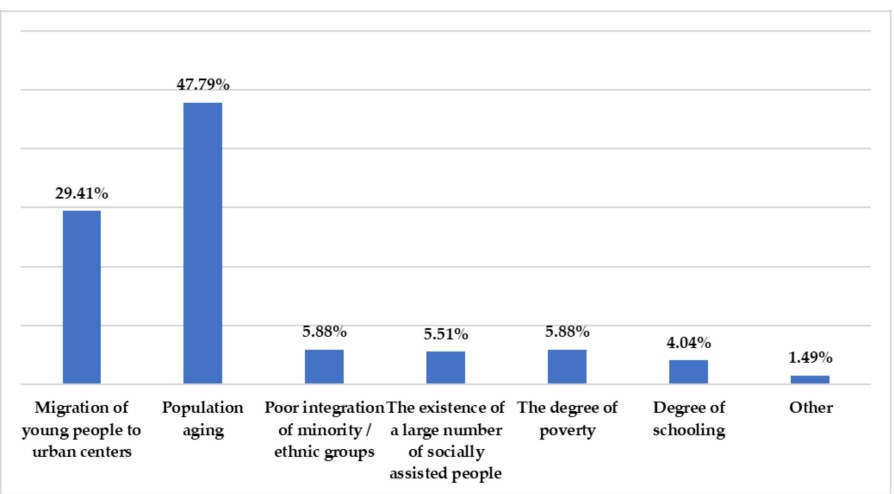

**Figure 2.** Analysis of the structure of the respondents according to the factor that influences the current stage of development of the locality they represent. Source: processing data obtained following the application of the questionnaire, applied between 15 January and 15 February 2021.

Analyzing the point of view of the representatives of the localities that answered the questionnaire regarding the main problem faced by the locality they manage, 49.26% of the respondents consider that the small number of jobs is the main problem. In the opinion of 20.31% of the respondents, the poor development of the urban infrastructure system represents the main problem faced by the locality they represent (Figure 3).

It is important to observe that in Figure 3, to a lesser extent, the representatives of the localities that responded to the questionnaire consider the poor development of medical offices, as well as the lack of leisure spaces as some of the main economic and social problems faced by the communities they represent (Figure 3).

The high share of local representatives who consider that the lack of jobs is the main problem of the commune can be explained by the fact that there is a significant share of agricultural business, which generates a limited number of jobs, often small and medium farms work the land with their own resources, without generating jobs at the local community level (Figure 3).

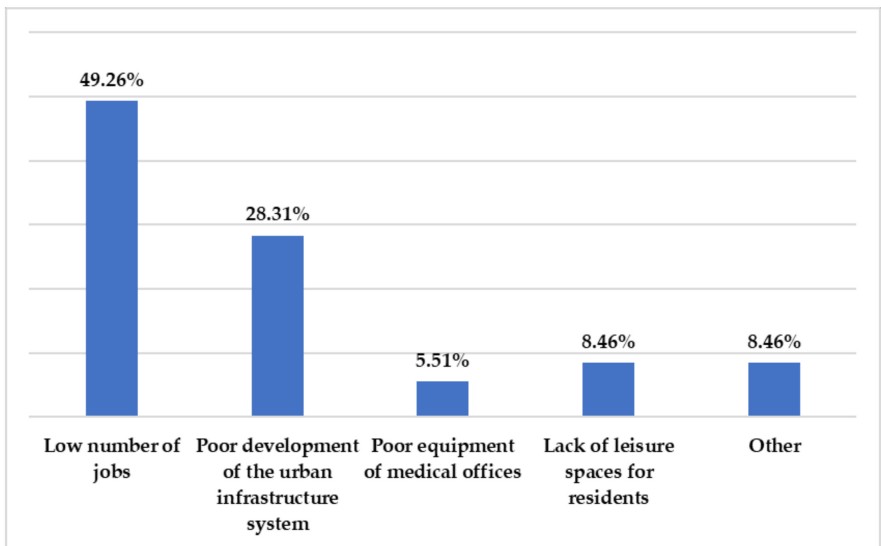

**Figure 3.** Analysis of the structure of the respondents regarding the main problem faced by the locality they represent. Source: processing data obtained following the application of the questionnaire, applied between 15 January and 15 February 2021.

According to the processed data about the main development perspective, towards which the locality they represent should be directed, 31.98% consider that the identification of a form of support to develop/establish an agricultural business is the main development perspective of the community they represent. Investments in infrastructure (water, gas, sewerage, roads, public lighting) are the main perspective of community development that it represents for 25.37% of respondents. For 26.84% of respondents, support for the development or establishment of non-agricultural businesses could be the main direction of development of the locality they represent. Only 11.40% of respondents believe that social investments should be the main objective of rural development for the locality they represent (Figure 4).

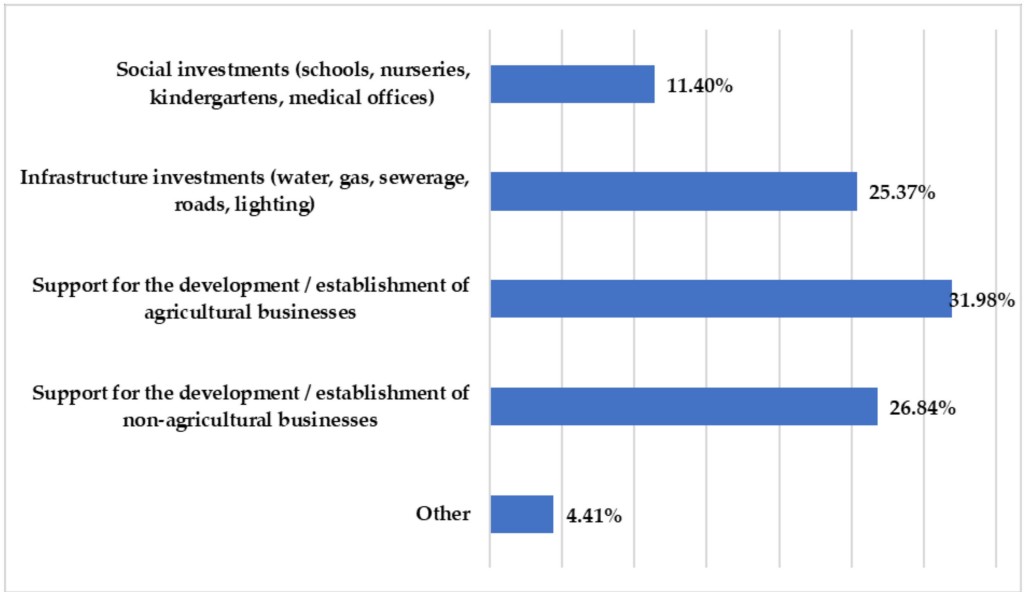

**Figure 4.** Analysis of the structure of the respondents regarding the development perspectives of the locality they represent. Source: processing data obtained following the application of the questionnaire, applied between 15 January and 15 February 2021.

Asked about the main impediment of accessing European funds, 39.34% of respondents believe that the lack of co-financing needed for investment is the main obstacle in accessing support measures with European funding for the development of the localities they represent. In the opinion of 23.16% of respondents, the excessively strict conditions to be met when accessing European funds may be the main barrier when accessing European funds, while bureaucracy may be an impediment in the opinion of 24.63% of respondents. A percentage of 8.82% of the total respondents do not consider such investments for the localities they represent. Analyzing the answers provided by the representatives of the localities, it can be observed that the lack of co-financing remains a topical issue at the level of rural communities. Additionally, the bureaucracy and excessively strict conditions determine a low degree of access to European funds for the development of rural communities in Romania (Figure 5).

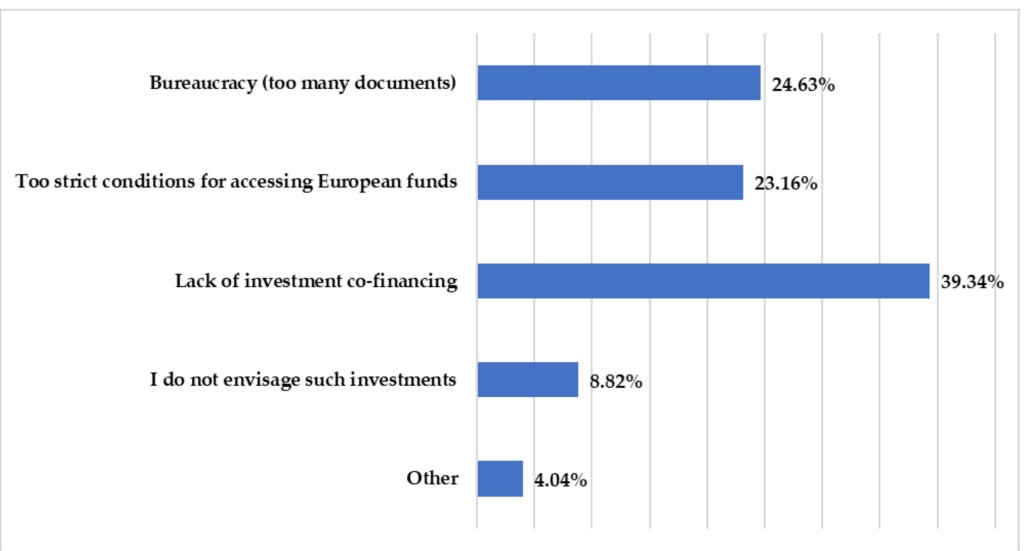

**Figure 5.** Analysis of the structure of the respondents regarding the factor that determines reduced investments through European funds, in the localities they represent. Source: processing data obtained following the application of the questionnaire, applied between 15 January and 15 February 2021.

For the analysis from a statistical point of view, the answers the data obtained were selected, in order to be described, only the questions that registered medium and strong links between the variables.

Analyzing the structure of the respondents classified according to the population of the commune they represent and the surface of the commune they manage, there can be observed an average association in intensity which is directly proportional (R = 0.440) (Table 1).

At the same time, the Chi-square value was calculated at 63.86, being higher than the critical value. The data analyzed in this way can lead to the conclusion that there is a very significant association between respondents representing localities with over 3000 inhabitants and those who manage an area of the commune of more than 100 km$^2$ (Table 1).

Following the processing and analysis of the data present in Table 2 it can be observed that regarding the structure of the respondents according to the dominant profile of the economic agents operating within the locality they represent and the main problem identified at the community level, there is an average association in intensity, being determined a value of R of 0.459 (Table 2).

**Table 1.** Analysis of the structure of the respondents classified according to the population and the area of the commune it administers.

| | | What Is the Area of the Commune You Manage? | | | Total |
|---|---|---|---|---|---|
| | | Under 50 km$^2$ | 50–100 km$^2$ | Over 100 km$^2$ | |
| What is the population of the commune you represent? | 501–1000 population | 15 | 4 | 4 | 23 |
| | 1001–2000 population | 20 | 27 | 4 | 51 |
| | 2001–3000 population | 23 | 28 | 32 | 83 |
| | Over 3000 population | 11 | 36 | 68 | 115 |
| Total | | 69 | 95 | 108 | 272 |
| Pearson Chi-Square | | | | 63.863 a | |
| Critical value ($p > 0.05$) | | | | 12.59 | |
| Cramer's V | | | | 0.343 | |
| Pearson's R | | | | 0.440 | |

Source: processing data obtained following the application of the questionnaire, applied between 15 January and 15 February 2021.

**Table 2.** Analysis of the structure of the respondents according to the dominant profile of the economic agents that operate within the locality they represent and the main problem identified at the community level.

| | | What Do You Consider to Be the Main Problem Identified at the Community Level? | | | | | | | |
|---|---|---|---|---|---|---|---|---|---|
| | | Young Migration | Population Aging | Poor Integration of Minority/Ethnic Groups | The Existence of a Large Number of Socially Assisted People | The Degree of Poverty | Degree of Schooling | Other | Total |
| The dominant profile of the economic agents (companies) within the locality you represent is. | Agricultural | 72 | 124 | 8 | 7 | 12 | 3 | 0 | 226 |
| | Non-agricultural | 8 | 6 | 8 | 8 | 4 | 8 | 4 | 46 |
| Total | | 80 | 130 | 16 | 15 | 16 | 11 | 4 | 272 |
| Pearson Chi-Square | | | | 88.120 a | | | | | |
| Critical value ($p > 0.05$) | | | | 12.59 | | | | | |
| Cramer's V | | | | 0.569 | | | | | |
| Pearson's R | | | | 0.459 | | | | | |

Source: processing data obtained following the application of the questionnaire, applied between 15 January and 15 February 2021.

Following the processing and analysis of the data highlighted in Table 2 can be seen that the Chi-square value is 88.12 and is higher than the critical value. Analyzing the results obtained, it can be observed that there is a very significant association between those who represent localities where the dominant profile of economic agents is agriculture and population aging (Table 2).

According to the data presented in Table 3 about the structure of the respondents according to the dominant profile of the economic agents operating within the locality they represent and their opinion on the types of funds that can contribute to the development of localities, an average association in intensity resulted, being determined a value of R of 0.38. Moreover, its Chi-square value of 47.47 indicates a very strong association between the dominant profile of agricultural economic agents and the sources of financing from the state budget used for the development of the localities (Table 3).

**Table 3.** Analysis of the structure of the respondents according to the dominant profile of the economic agents operating within the locality they represent and their opinion on the types of funds that can contribute to the development of the localities.

| | | What Do You Think Could Be the Most Important Funds That Contribute to the Development of the Localities You Manage? | | | | Total |
|---|---|---|---|---|---|---|
| | | Own Funds | From the State Budget | European Union Funding | Other | |
| The dominant profile of the economic agents (firms) within the locality you represent is: | Agricultural | 76 | 126 | 16 | 8 | 226 |
| | Non-agricultural | 4 | 18 | 16 | 8 | 46 |
| Total | | 80 | 144 | 32 | 16 | 272 |
| Pearson Chi-Square | | 47.472 a | | | | |
| Critical value ($p > 0.05$) | | 7.81 | | | | |
| Cramer's V | | 0.42 | | | | |
| Pearson's R | | 0.38 | | | | |

Source: processing data obtained following the application of the questionnaire, applied between 15 January and 15 February 2021.

Following the processing and analysis of the data highlighted in Table 4 about the structure of respondents according to their opinion on the main development perspective of the locality and knowledge of information aimed at accessing European funds, there is an average association in intensity, being determined a value of R of −0.450 (Table 4).

**Table 4.** Analysis of the structure of the respondents according to their opinion regarding the main development perspective of the locality and knowledge of the information related to accessing European funds.

| | | Are You Aware of the Information Aimed at Accessing European Funds? | | Total |
|---|---|---|---|---|
| | | Yes | No | |
| What do you think it should be the main development perspective that the locality you represent should be heading towards? | Infrastructure investments | 4 | 65 | 69 |
| | Social investments | 12 | 19 | 31 |
| | Development/establishment support agricultural business | 68 | 19 | 87 |
| | Development/establishment support non-agricultural business | 36 | 37 | 73 |
| | Other | 12 | 0 | 12 |
| Total | | 132 | 140 | 272 |
| Pearson Chi-Square | | 94.966 a | | |
| Critical value ($p > 0.05$) | | 9.49 | | |
| Cramer's V | | 0.591 | | |
| Pearson's R | | −0.450 | | |

Source: processing data obtained following the application of the questionnaire, applied between 15 January and 15 February 2021.

Additionally, the Chi-square value of 94.97 indicates that there is a very strong association between the support needed for the development/establishment of agricultural businesses and the knowledge of information aimed at accessing European funds (Table 4).

It is important to observe that in Table 5, the structure of the respondents according to their opinion on the main development perspective of the locality and accessing European funds for the localities they represent, there is an average association in intensity, being determined an R-value of −392 (inversely proportional). Additionally, the Chi-square

value of 73.82 indicates that there is a very strong association between support for the development/establishment of agricultural affairs and access to European funds (Table 5).

**Table 5.** Analysis of the structure of the respondents according to their opinion on the main development perspective of the locality and access to European funds for the localities they represent.

| | | Have You Accessed European Funds for the Development of the Locality You Represent? | | Total |
|---|---|---|---|---|
| | | Yes | No | |
| What do you think should be the main development perspective that the locality you represent should be heading towards? | Infrastructure investments | 0 | 69 | 69 |
| | Social investments | 8 | 23 | 31 |
| | Development/establishment support agricultural business | 56 | 31 | 87 |
| | Development/establishment support non-agricultural business | 36 | 37 | 73 |
| | Other | 4 | 8 | 12 |
| Total | | 104 | 168 | 272 |
| Pearson Chi-Square | | 73.817 a | | |
| Critical value ($p > 0.05$) | | 9.49 | | |
| Cramer's V | | 0.521 | | |
| Pearson's R | | −0.392 | | |

Source: processing data obtained following the application of the questionnaire, applied between 15 January and 15 February 2021.

Analyzing the data from Table 6 about the structure of the respondents according to their opinion on the knowledge of information aimed at accessing European funds and accessing European funds for the localities they represent, it is noted, a strong association in intensity, being determined a value of R of 0.810 (Table 6).

**Table 6.** Analysis of the structure of the respondents according to their opinion on the knowledge of the information about accessing European funds for the localities they represent.

| | | Have You Accessed European Funds for the Development of the Locality You Represent? | | Total |
|---|---|---|---|---|
| | | Yes | No | |
| Are you aware of the information aimed at accessing European funds? | Yes | 104 | 28 | 132 |
| | No | 0 | 140 | 140 |
| Total | | 104 | 168 | 272 |
| Pearson Chi-Square | | 178.586 a | | |
| Critical value ($p > 0.05$) | | 3.841 | | |
| Cramer's V | | 0.810 | | |
| Pearson's R | | 0.810 | | |

Source: processing data obtained following the application of the questionnaire, applied between 15 January and 15 February 2021.

Additionally, the Chi-square value of 178.59 indicates that there is a very strong association between those who are aware of the information on European funds and those who have accessed European funds (Table 6).

Following the processing and analysis of the data presented in Table 7 about the structure of the respondents according to their opinion on the knowledge of information regarding access to European funds and their opinion on the main impediment to accessing

European funds, it can be observed an average association in intensity, being determined a value of R of 0.429 (Table 7).

**Table 7.** Analysis of the structure of the respondents according to their opinion on the knowledge of the information aimed at accessing European funds and their opinion on the main impediment related to accessing European funds.

| | | What Do You Consider to Be the Main Impediment to Accessing European Funds? | | | | | |
|---|---|---|---|---|---|---|---|
| | | Bureaucracy | Too Strict Conditions Required for Access European Funds | Lack of Investment Co-Financing | I Do Not Envisage Such Investments | Other | Total |
| Are you aware of the information aimed at accessing European funds? | Yes | 52 | 48 | 20 | 8 | 4 | 132 |
| | No | 15 | 15 | 87 | 16 | 7 | 140 |
| Total | | 67 | 63 | 107 | 24 | 11 | 272 |
| Pearson Chi-Square | | | | | | | 82.993 a |
| Critical value ($p > 0.05$) | | | | | | | 9.487 |
| Cramer's V | | | | | | | 0.552 |
| Pearson's R | | | | | | | 0.429 |

Source: processing data obtained following the application of the questionnaire, applied between 15 January and 15 February 2021.

Additionally, the Chi-square value of 82.99, Indicates that there is a very strong association between those who are not aware of the information on European funds and the lack of co-financing of investments. In other words, those who are not aware of the information on accessing European funds face problems related to co-financing the investment (Table 7).

Following the processing and analysis of the data about the structure of the respondents according to their opinion on accessing European funds and their opinion on the main impediment related to accessing European funds, there is an average association in intensity, being determined a value of R of 0.55 (Table 8).

**Table 8.** Analysis of the structure of the respondents according to their opinion on accessing European funds and their opinion on the main impediment related to accessing European funds.

| | | What Do You Consider to Be the Main Impediment to Accessing European Funds? | | | | | |
|---|---|---|---|---|---|---|---|
| | | Bureaucracy | Too Strict Conditions Required for Access European Funds | Lack of Investment Co-Financing | I Do Not Envisage Such Investments | Other | Total |
| Have you accessed European funds for the development of the locality you represent? | Yes | 48 | 40 | 16 | 0 | 0 | 104 |
| | No | 19 | 23 | 91 | 24 | 11 | 168 |
| Total | | 67 | 63 | 107 | 24 | 11 | 272 |
| Pearson Chi-Square | | | | | | | 94.905 a |
| Critical value ($p > 0.05$) | | | | | | | 9.487 |
| Cramer's V | | | | | | | 0.59 |
| Pearson's R | | | | | | | 0.55 |

Source: processing data obtained following the application of the questionnaire, applied between 15 January and 15 February 2021.

With a Chi-square value of 94.90 which is higher than the critical value, it can be stated that there is a very significant association between those who did not access European funds and the lack of co-financing of the investment (Table 8).

*4.2. What Solutions Can Contribute to the Revitalization of Rural Areas?*

Analyzing the data obtained from the application of the questionnaire to the representatives of the local authorities, allowed the identification of three major problems faced by the rural localities: the aging population, the migration of young people to urban centers and the lack of jobs.

Starting from the identification of these three main problems faced by rural authorities, the following are solutions to reduce the migration of young people from rural areas to urban centers, by creating jobs and integrating them back into the community. Job creation will be achieved with the help of a digital tool to which both local authorities and potential investors would have access. This digital tool will gather centralized data about each region and allow investors to discover the potential of each region and find the most suitable area to grow their business taking into account the resources found in the area. Additionally, the academic environment has an essential role in this process, thinking and implementing study programs for young people according to the specific needs of rural areas.

Government authorities can also contribute through financial investments, exactly in the regions where these investments are needed, taking into account the specifics (what investments need to be made) and how urgent they are.

### 4.2.1. The Role of Young People in Local Communities

European funds can be a way for rural communities to develop harmoniously and, where young people can start families if the living conditions are good and very good.

Indeed, submitting a project can be a difficult thing, starting from identifying programs/measures that can be accessed according to the needs and availability of the commune, to bureaucracy and preparation of studies to be attached to the funding file, so local authorities need specialized and well-trained people.

According to Figure 6, young people from rural areas should be encouraged, to study, to go to the faculties that are located in the big cities, in order to train and accumulate the necessary knowledge. If they come from families without material means, the state should offer them financial support so that they can continue their studies, offering them on their return home, a place in the town halls, where they can offer back to the community their knowledge. They can also promote funding measures among residents and support them in accessing European funds for their own activity.

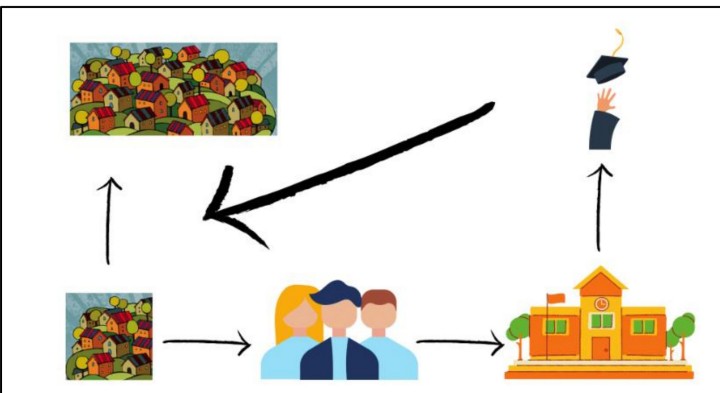

**Figure 6.** Flow for the integration of young people in the rural communities from which they come. Source: in-house development.

### 4.2.2. Digital Platform for Monitoring the Rural Area

Digitization is a new area introduced in the future of the CAP, so this approach must be implemented in areas where there will be no repercussions (for example, repercussions on jobs, regardless of the field of activity).

Regarding the field of rural development, it is necessary to develop a platform to monitor the situation of rural localities. This can be carried out by creating a platform, to which will have access: the academic environment, decision-making institutions (MADR, County Departments, etc.), potential investors. Analyzing the data that will be found in the digital platform will make it possible to identify current issues and develop specific and differentiated measures from one region to another.

The data that will be found in the platform will be able to be collected from rural localities, the data can be centralized and can be accessed by stakeholders. The platform will integrate data about: population, birth rate, mortality, economic agents classified according to profile, infrastructure, sales markets, distances from sales centers, tourist potential (including agrotourism), as well as contact details of localities (Figure 7).

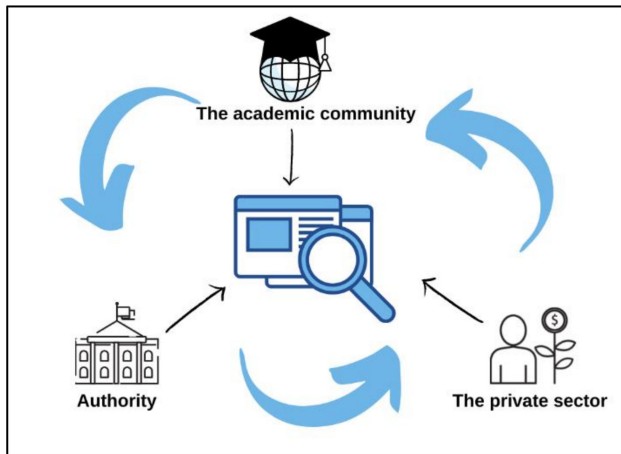

**Figure 7.** Rural monitoring platform. Source: in-house development.

Through this platform, analyzing the available data the authorities will be able to identify problems at the micro-regional level so that they can make the best decisions from a socio-economic point of view.

The private environment that wants to make investments in rural areas can take decisions based on accurate data (taking into account the share of labor, existing infrastructure, etc.). The academic environment will have access to these data and can carry out specialized studies that can support the authorities and economic agencies.

### 4.2.3. Model of Sustainable Development of Rural Localities

Based on the research presented previously, a series of measures were developed that can contribute to the sustainable development of rural localities:

1. For local authorities

   - Development of municipal services, with emphasis on the creation/modernization of medical offices, kindergartens, schools, drinking water, gas, sewerage, lighting. This can be achieved through European funds and by creating a compartment at the city hall level to deal with identifying and attracting these funds. This is an important component from the rural localities that already show an improvement in living conditions and the attraction of the population to these localities.
   - Support for private companies by making available land on the outskirts of localities, and exemption from paying local taxes, for those legal entities that want

to invest in rural areas. This will lead to the creation of jobs for the population in the area, generating funds for the local budget.

- Land for building houses, for young families who want to start a family in rural areas.
- Supporting farmers to access European funds, given that this is easier.
- Supporting farmers to adhere to associative forms, which will lead to the development of agricultural holdings from all points of view.
- Creating local centers for the conditioning, processing and marketing of agricultural products.
- Increasing taxes for homeowners in a state of obvious degradation and offering the possibility of renting for disadvantaged people.
- The activities related to the care of the elderly in the localities should be carried out by the socially assisted persons, but also their use for the good management of the commune.
- Using the platform mentioned above, to identify problems at a regional and common level.

2. For the academic environment

- Establishing partnerships between local authorities, through vocational schools and specialized units, through which young people should be aware of the importance of continuing their studies in the fields they consider important. Subsequently, returned to the localities from which they come, they can contribute to the development of these communes.
- Students to do internships on farms in these areas, and to observe and identify solutions that can contribute to the development of localities where they practice (faculties specializing in "rural development").
- Providing scholarships for students from disadvantaged families, who, although they have a good intellectual capacity, cannot continue their studies due to lack of income.

3. For the private environment

- Carrying out internships for pupils/students, which have a double advantage. On the one hand, pupils/students can find out if that specialization is to their liking, and companies can identify young people who are eager to assert themselves.
- Granting scholarships to pupils/students with good results in education provided that after completing their studies to be employed in companies.

## 5. Conclusions, Implications and Limitations

Rural areas as well as their sustainable development are a concern for many researchers, who try to come up with various solutions, from creating models for the development of rural tourism (non-agricultural activities) to the analysis of rural development policies and the role of local action groups in regional development. All these solutions must take into account the main actors (stakeholders) in these areas (local authorities, economic agents and inhabitants).

Analyzing the data, it can be observed that most of the respondents (40.07%) represent localities that have more than three villages. The significant share of respondents who represent localities with more than three villages can be explained by the fact that, at the national level, on average, a commune consists of five villages.

Processing and analyzing the data obtained, it can be noted that 39.71% of the respondents represent localities with an area of over 100 km, 34.93% of the respondents represent localities with an area between 50 and 100 km, while only 25.27% of respondents represent localities that have an area of less than 50 km

Analyzing the data, in the case of 83.09% of the respondents, the dominant profile of the economic agents from the locality they represent is the agricultural one, and only in

the case of 16.91% of the respondents, the dominant profile of the locality they represent is non-agricultural. The significant share of agricultural business in the localities that the respondents of the questionnaire represent shows that most of the resources of the inhabitants are concentrated on agricultural activity, this being the main source of income. Encouraging rural entrepreneurship in other areas of activity can be a source of income and better use of the natural resources of the respondents' localities.

Processing and analyzing the data obtained, it can be observed that for most of the respondents (47.79%) the aging population is the main problem of the locality they represent. A percentage of 29.41% of the respondents consider that the migration of young people to urban centers could be the main problem faced by the locality they represent.

Regarding the problems faced by rural communities, most of the respondents consider that the low number of jobs is the main problem of the locality they represent. The poor development of the urban infrastructure system is the main problem in the opinion of 28.31% of respondents.

Regarding the rural development strategy of the locality they represent, a significant percentage of respondents believe that resources should be directed towards identifying forms of support to develop/set up agricultural businesses. Additionally, investments in infrastructure (water, gas, sewerage, roads, public lighting) could be the main development perspective of the localities that respondents represent. For 26.84% of the respondents, the support for the development or establishment of non-agricultural businesses could represent the main direction of development of the locality they represent. Only 11.40% of respondents believe that social investments should be the main rural development objective of the locality they represent.

European funds can play an important role in the development process of rural communities. Access to information about the available support measures for rural development and consolidation is extremely important. Asked about the extent to which they know the information on accessing European funds, most of the respondents are not aware of the process of accessing European funds.

Only 38.24% of the total respondents claim that they benefited from European funds for the development of the localities they represent.

According to respondents, the lack of co-financing of investment is the main obstacle in accessing European-funded support measures. Additionally, the strict conditions that the community must meet when accessing European funds, but also the bureaucracy can be barriers in accessing European funds for the localities they represent.

Over 50% of respondents believe that the funds needed for the development of the localities they represent should come from the state budget.

The process of accessing European funds is not an easy one, and the multitude of support measures for the rural environment requires the existence, at the level of local administration, of such a person specialized in the process of accessing European funds. Only 7.35% of respondents inform they have, at the level of local government, such a person responsible for accessing European funds.

Representatives of local authorities are aware that the aging of the population and the migration of young people from the administered areas prevent their development, but the measures taken are almost non-existent. At the same time, the representatives of the local authorities know the reasons why young people leave these localities, which is also highlighted by the significant share of 49.26% (Figure 3) which indicates that the main reason would be the lack of jobs, followed poor infrastructure development.

The representatives are aware of the measures that need to be taken at the locality level, in order to solve the problems that they have observed, starting from the support for the development of agricultural and non-agricultural affairs, to the investments in infrastructure (water, gas, etc.), but cites reasons related to the lack of co-financing of rural investments, bureaucracy or too strict conditions for accessing European funds.

In conclusion, according to the analyzed data, it can be stated that the representatives of the local authorities know the problems they face and the solutions to solve these

problems, but, in practice, things remain unchanged. This can be attributed to insufficient funds that go to the local budget, which can not cover the investments, and are used only for the commune to "survive", without prospects and horizons. Additionally, according to the analyzed data another reason is the social factor, in which the population is aging and young people choose to migrate to urban centers in search of jobs. Without diversified economic activities, which are not mainly concentrated in agriculture, the jobs available in rural areas are insufficient to ensure a decent living for the inhabitants of these areas. Young people go to study in cities, and rural areas are left without young entrepreneurs. In this sense, it is necessary to create a synergy between local authorities, researchers and the private sector represented by potential investors. This solution can also be accessed by government authorities, which can allocate investment funds according to regional needs. In order to address these issues, the government should provide funding to local authorities, depending on the shortcomings identified through the digital tool, by acting precisely on the aspects that hinder the development of those areas.

**Author Contributions:** The authors worked together for this research, but, per structure, conceptualization V.C.T., M.M.M. and A.U., methodology, software validation and resources, V.C.T., M.M.M. and A.U., formal analysis, A.U. and E.A.D. writing—original draft preparation, and writing—review and editing, V.C.T. and E.A.D. All authors have read and agreed to the published version of the manuscript.

**Funding:** This research received no external funding.

**Institutional Review Board Statement:** Not applicable.

**Informed Consent Statement:** Not applicable.

**Data Availability Statement:** The data presented in this study are available on request from the corresponding author. The data are not publicly available due to privacy.

**Conflicts of Interest:** The authors declare no conflict of interest.

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
