# Peer review of "Sustainable Development of the Rural Areas from Romania: Development of a Digital Tool to Generate Adapted Solutions at Local Level"

_sustainability, doi:10.3390/su132111921_

Round 1

Reviewer 1 Report

General comments: awkwardly worded in English, needs significant stylistic revision

Title: what is the digital solution? Is this referring to the questionnaire? Isn't navigation of the process/procurement of EU funding deemed the solution?

Awkward title: Reword to "Sustainable Development in Rural Romania: Development of a Digital Instrument to Analyze Causation and Formulate Local-Level Solutions "?  

I would reword the abstract as follows:

Rural Romania faces many problems, chief among them depopulation. In general, young adults frequently choose to migrate to urban centers or leave the country entirely to satisfy economic and/or social aspirations. The aim of this study is to identify intractable issues that hinder rural development, and to identify actionable solutions. In this regard, a questionnaire was developed and administered to local authorities. Results were analyzed with the SPSS Statistical Program, with the Pearson Chi-square, Cramer and Pearson coefficients determined. [Results were...] [General recommendations based on results]

General Formatting: After printing pdf of manuscript, bottom line of many pages (lines 96, 149, 189, 404, 457) are truncated. Figure 1, as well as Tables 3 and 5 are similarly cut off.

Lines 6 and 7: use official affiliation emails rather than yahoo.com?

Abstract, line 18: provide a summation of your findings/conclusions.

Keywords, line 19: use more descriptive alternative to “local authorities”, perhaps localities or local/regional governments?

44: what specific health crises?

48: does “crafts” imply practices (which is mentioned later)? What is the agricultural context?

53 and 54: non-ag businesses implies “multiplier effect”?

61-62: include illustrative depiction of sustainability?

76: peri-urban areas?

81: reword to “Our aim is to identify…”

84: “According to Darnhofer…” – uses parenthetical citation style while rest are numeric and bracketed, e.g. [26]

91: same as above – reword to say “It has been posited that XYZ [number].” Instead of mentioning authors and paper titles by name in the narrative

100: same as above

107: same as above

112: same as above

132: what value or benchmark constitutes a “representative number” of respondents?

Show map of area(s) sampled, as well as economic zones mentioned earlier?

147: “electronic format” = digitally

369: “own processing” = in-house development?

Author Response

Thank you for your suggestions, which I appreciate and consider extremely constructive for the improvement of the work.

Reviewer 2 Report

See attached

Author Response

(The authors gave the same response as above.)

Reviewer 3 Report

This is an interesting and insightful study regarding issues that hinders the development in Romanian rural regions. The manuscript is concise and well-written. The experiment design and data analysis are simple but straightforward. I only have a few suggestions.

  1. The title of the manuscript is too general and lengthy. The use of “research” is redundant since it does not add extra values in helping readers understanding the study. Maybe have a title like “Questionnaire-based problem identification for Romanian rural areas…”.
  2. Abstract is not properly written. It does not justify the need for the study. It is too general (see my next comment). It does not summarize the findings and conclusions of the study.
  3. The authors used “problems”, “developments” and “solutions” frequently throughout the manuscript. The words themselves are very general. The authors should specify what types of “problems”, “developments” and “solutions” are being identified in the study. Economic? Environmental? Cultural? Proper adjustments should be made throughout the manuscript from title to conclusion.
  4. Explain the necessity of the study and the knowledge gap in current literature to help readers understand why the study was conducted.
  5. Maybe have a dedicated subsection under/beside the Methodology section showing the original questionnaire and the 12 questions.
  6. Much of the section 5 “Conclusions, Implications and Limitations” is repeating the information from previous sections. Consider rewriting and keeping it concise and more focused.

Author Response

(The authors gave the same response as above.)

Reviewer 4 Report

Dear authors:

Thank you for your paper on rural development in Romania. The topic of the paper is certainly very important and you present interesting survey data. However, I see several shortcomings in the manuscript. There is, however, potential in your research and after building a more coherent argument, it might be suitable to submit it again. However, the modifications necessary are too substantial to recommend revisions based on the current version. Here is a summary of my points:

  1. I miss a clear scope. While you aim to identify factors hindering development in rural Romania, it is unclear what your specific research question is.
  2. I do not understand why you summarize four studies on the Austrian context in your section on the theoretical background. In that section, you could perhaps refer to a model of rural development. But that would require a more precise focus of your overall paper.
  3. How do your hypotheses emerge from the existing literature and your theoretical framework?
  4. While you present comprehensive and interesting data, you do not present a convincing analysis of the data. What do we learn from the data? There should be an analysis or discussion section on this.
  5. It is unclear to me how section 4.2 relates to your data. Moreover, the recommendations you provide would need to be sustantiated by your analysis and discussion, which is not the case.
  6. One minor comment: You claim your sample is representative. I would recommend providing the calculations on this.

As I said, I think the paper's topic and data are in principle very interesting and would encourage you to use this in a new or substantially revised paper.

Author Response

(The authors gave the same response as above.)

Round 2

Reviewer 1 Report

Line 10: change correspondence email to conform with email listed previously

Line 40: "rep-resented" spelling?

Author Response

Thank you for your suggestions. We've fixed your comments.

Thank you again for the feedback you have provided, you have helped to improve the work.

Round 3

Reviewer 1 Report

Content/empiricism is fine, and the readability of the narrative is improved, but still could use further editing by a native English speaker.

Author Response

Thank you for your support. I would like to inform you that I have checked the translation of the paper into English and corrected it where appropriate, according to your suggestions. Also, at the email address of the correspondent, I passed the address of the institution (which I omitted in previous reviews). This correction does not change the structure of the authors as it was originally sent.